Microbiology **Spectrum**

∂ | **Open Peer Review** | Clinical Microbiology | Research Article

# The role of L-type amino acid transporter 1 in the pathogenicity of *Candida albicans*

Yibing Lan,[1] Cheng Wu,[1] Peng Du,[2] Jie Jiao,[3] Chunming Li,[1] Ketan Chu,[1] Tao Zhang,[1] Peiqiong Chen,[1] An Li,[1] Wenxian Xu,[1] Xinyi Ying,[3] Jianhong Zhou,[1] Wenlong Lin,[4] Linjuan Ma,[1] Yizhou Huang[1]

**ABSTRACT** Our recent study showed that *lat1*, which encodes L-type amino acid transporter 1 (Lat1) was centrally positioned in the gene co-expression network of a *Candida albicans* (*C. albicans*) infection model of cell culture and suggested that this gene might play a critical role in the pathogenesis of *C. albicans* infections. In the current work, we used CRISPR-Cas9 to generate a *C. albicans lat1* mutant strain (*lat1Δ/Δ*), using *C. albicans* SC5314 as the wild-type (WT) strain. An *in vitro Candida* infection model using vaginal epithelial cells and a murine model of vulvovaginal candidiasis was used to investigate the role of *lat1* in the pathogenicity of *C. albicans*. It was found that *lat1* played important roles in cell proliferation, morphogenesis, early adherence, and biofilm formation of *C. albicans*. VK2-E6E7 human vaginal epithelial cells infected with the *lat1Δ/Δ* mutant strain also showed significantly lower activation of Toll-like receptor 2/4 (TLR2/4) and the downstream MyD88/NF-κB signaling pathway, accompanied by an attenuated secretion of inflammatory cytokines. In the murine model of vulvovaginal candidiasis, infection caused by *lat1Δ/Δ* showed a lower fungal burden in the vaginal lavage fluid, reduced production of inflammatory cytokines, and diminished recruitment of neutrophils to the vaginal epithelium, relative to that caused by the WT strain. Based on these findings, we conclude that *lat1* plays an important role in the host-pathogen interactions in *C. albicans* infections by impacting the virulence of *C. albicans* and host inflammatory responses; the latter was possibly via the TLR2/4-MyD88-NF-κB pathway.

**IMPORTANCE** Effective management of vulvovaginal candidiasis requires a comprehensive understanding of virulence genes in *C. albicans* and their role in the host-pathogen interactions. Our study identified *lat1* as an important virulence factor of *C. albicans*, impacting fungal adaptations, including morphogenesis and biofilm formation, virulence, and host immune responses, such as the epithelial TLR2/4-MyD88-NF-κB activation in the murine model of vulvovaginal candidiasis. Lat1-mediated amino acid transport may serve as a key metabolic constraint during early infection. Further research is warranted to validate the clinical importance of *lat1* in vulvovaginal candidiasis and its potential as a therapeutic target.

**KEYWORDS** vulvovaginal candidiasis, *Candida albicans*, fungal virulence, host immune response, *lat1*

**Peer Reviewer** Yue Qu, Monash University, Melbourne, Australia

Address correspondence to Yizhou Huang, huangyz1219@zju.edu.cn, or Linjuan Ma, mlj8998@126.com.

Yibing Lan and Cheng Wu contributed equally to this article. Author order was determined both last name alphabetically and in order of increasing seniority.

The authors declare no conflict of interest.

See the funding table on p. 13.

Vulvovaginal candidiasis (VVC) is a common fungal infection of the female lower genital tract, with *C. albicans* being the dominating pathogen (1, 2). It has been reported that approximately 75% of women of childbearing age had at least one episode of VVC, and 5–10% of these women may even experience recurrent vulvovaginal candidiasis (RVVC), defined as at least three episodes of infection within a 12-month period (3–6). Although being non-fatal, VVC adversely impacts the quality of life and mental health of affected women (7–10). In recent years, the incidence of VVC has shown

a clear upward trend (2, 4), possibly due to other confounding factors, such as the widespread use of antibiotics, corticosteroids, immunosuppressants, chemoradiotherapy, and chemotherapy.

*C. albicans* may reside in the human vagina as a commensal organism, typically at low abundance and without evident symptoms. It has been reported to play important roles in maintaining the ecological balance of the vaginal microbiota and in the self-cleaning processes of the vagina (7, 11). When the balanced vaginal microenvironment is disrupted, *C. albicans* may multiply and invade the vaginal epithelium(12), triggering a cascade of vaginitis symptoms (13). The host-pathogen interactions in the vagina, and their key genes, are still poorly understood.

L-type amino acid transporter 1 (Lat1) is an important member of the LAT family, which is mainly responsible for transporting large neutral branched-chain amino acids and aromatic amino acids (14). The importance of *lat1* in *C. albicans* and some other fungal pathogens has not been fully elucidated (15). In our very recent study (16), gene expression profiling of *C. albicans*-infected HeLa cells suggested that *lat1* occupied a relatively central node in the co-expression network of pathogenicity-related genes, implicating a possibly important role that *lat1* plays in *C. albicans* infection. Nonetheless, direct evidence supporting the importance of *lat1* in the pathogenicity of *C. albicans* has not been reported.

Toll-like receptors (TLRs) are key pattern-recognition receptors that detect pathogen-associated molecular patterns (PAMPs) and initiate host immune responses (17). In mammals, most TLRs signal through the MyD88-dependent pathway, leading to the activation of nuclear factor kappa B (NF-κB) (18). Upon TLR activation, a signaling cascade results in phosphorylation and degradation of IκBα and subsequent nuclear translocation of NF-κB, which in turn upregulates numerous cytokines, chemokines, and inflammatory mediators (19). This TLR/MyD88/NF-κB pathway plays a pivotal role in innate immune responses and has been implicated in the pathogenesis of VVC(20). However, it is not fully understood how fungal factors modulate this pathway during *C. albicans* infection of the vaginal mucosa.

In this study, we aimed to understand the detailed roles that *lat1* plays in the host-*C. albicans* interactions and to identify signaling pathways that might be involved in *C. albicans*-induced vaginal inflammation. We constructed a *C. albicans lat1Δ/Δ* mutant strain using CRISPR-Cas9 and assessed its virulence phenotypes using *in vitro* and *in vivo* models of vaginal *Candida* infection. We found that deletion of *lat1* significantly impaired *C. albicans* proliferation, morphogenesis, and adhesion and was associated with a reduced inflammatory response in the host. Our results support that *lat1* is an important virulence determinant of *C. albicans* and may influence the severity of *Candida* vaginitis through the TLR2/4-MyD88-NF-κB signaling pathway in vaginal epithelial cells.

## MATERIALS AND METHODS

### *C. albicans* strain and CRISPR-Cas9

*C. albicans* SC5314 (ATCC MYA-2876) was used as the wild-type strain. Yeast cells were grown overnight in YPD (2% D-glucose, 2% peptone, 1% yeast extract) broth at 30°C with shaking at 180 rpm. Cells were harvested by centrifugation at $2,500 \times g$, washed three times with phosphate-buffered saline (PBS), and resuspended in fresh broth medium for subsequent experiments.

The *lat1* gene was inactivated using a CRISPR-Cas9-mediated genome editing approach (21). Briefly, single guide RNAs (sgRNAs) targeting the *lat1* gene were designed, and a repair oligonucleotide was used as donor DNA to introduce two in-frame stop codons into the *lat1* open reading frame. Specifically, the endogenous sequence TTTGCC was replaced by TAATAG that encodes the stop codons TAA and TAG. Additionally, an EcoRI restriction site was introduced immediately downstream to facilitate screening. The CRISPR-Cas9 plasmid and repair oligonucleotide were co-transformed into *C. albicans* SC5314. After transformation, cells were plated on YPD agar plates, and putative *lat1*

mutants were identified by PCR and EcoRI digestion. Correctly edited clones were further confirmed by sequencing the modified *lat1* locus, and a homozygous loss-of-function strain was obtained and designated *lat1Δ/Δ*.

## *Candida* proliferation and yeast-hyphae morphogenesis assessments

Overnight cultures of *C. albicans* SC5314 (WT) and its *lat1Δ/Δ* mutant strain were harvested by centrifugation ($2,500 \times g$, 5 min), washed three times with PBS, and resuspended in 10 mL of fresh YPD broth to an initial optical density at 600 nm ($OD_{600}$) of 0.1. Cultures were subsequently incubated at 30°C with shaking at 200 rpm. Aliquots were collected at 1, 3, 6, 12, and 24 h, and the $OD_{600}$ at each time point was measured using a Cary series UV-Vis spectrophotometer (Agilent Technologies, USA).

Yeast-hyphae morphogenesis of *C. albicans* was assessed using a broth filamentation assay as previously described (22, 23). Briefly, overnight *C. albicans* cultures were adjusted to $3 \times 10^7$ cells/mL using a hemocytometer. One milliliter of the fungal suspension was added to 10 mL of YPD broth supplemented with 10% fetal bovine serum (FBS) and incubated at 37°C with shaking at 200 rpm. Samples were taken from the suspensions at 1 h, 3 h, and 6 h, washed with PBS, and immediately imaged with a light microscope (Leica DM6 B, Germany) at 400× magnification. Ten fields of view per slide were randomly selected, and the proportion of germ-tube-forming cells was determined.

## Cell culture

The human vaginal epithelial cell line VK2/E6E7 (ATCC CRL-2616) was used to study host-*Candida* interactions *in vitro*. Cells were cultured in keratinocyte serum-free medium (KSFM, Gibco, Grand Island, NY) supplemented with 5 ng/mL of recombinant epidermal growth factor, 50 µg/mL of bovine pituitary extract (Invitrogen, Grand Island, NY), and 100 units/mL of penicillin and streptomycin, respectively (Life Technologies, Grand Island). The culture was maintained at 37°C in a humidified incubator with 5% CO2.

## *In vitro* co-culture and adhesion assay

To assess adhesion capacity, overnight cultures of *C. albicans* SC5314 (WT) and the *lat1Δ/Δ* mutant strain were grown in YPD broth at 30°C. Cells were harvested, washed three times with PBS, resuspended in PBS, and adjusted to a concentration of $3 \times 10^7$ cells/mL using a hemocytometer. VK2/E6E7 cells in the logarithmic growth phase were detached with 0.25% trypsin, washed with PBS, and resuspended in antibiotic-free K-SFM. The cell suspension was adjusted to $1 \times 10^5$ cells/mL, seeded into 6-well plates, and incubated overnight at 37°C with 5% $CO_2$. Subsequently, 200 µL of the prepared *C. albicans* suspension was added to the VK2/E6E7 cells in the 6-well plates, and the co-cultures were incubated at 37°C with 5% $CO_2$ for 2 h. The supernatant was removed, and the wells were washed three times with PBS to eliminate non-adherent fungi. Cells were then fixed with 4% paraformaldehyde. For immunofluorescence labeling, VK2/E6E7 cells were stained with Phalloidin-Alexa Fluor 568 (A12380, Invitrogen, Thermo Fisher, USA), and *C. albicans* cells were labeled using an anti-*Candida albicans* antibody (ab53891, Abcam, USA), followed by incubation with a goat anti-Rabbit IgG H&L Alexa Fluor 488 secondary antibody (111-545-003, Jackson ImmunoResearch, USA). Images were acquired using a confocal laser scanning microscope (Fv1000, Olympus , Japan). The ratio of *C. albicans* to epithelial cells was determined as a measure of adhesion capacity.

## *In vitro* and *in vivo* cytokine quantification by ELISA

After 24 h of co-culture of VK2/E6E7 cells with *C. albicans* WT or *lat1Δ/Δ* strains in 6-well plates, culture supernatants were collected and centrifuged at 4°C to remove cellular debris. The supernatants were used to quantify TNF-α, IL-1β, IL-6, IL-10, and CCL2 by ELISA (BD Biosciences, USA), following the manufacturer's instructions.

For *in vivo* cytokine quantification, vaginal lavage samples were collected at the indicated time points post-inoculation and centrifuged at 4°C to remove debris. The supernatants were diluted 1:20 or at higher dilution factors in the assay buffer and analyzed for TNF-α, IL-1β, IL-6, IL-10, and CCL2 using ELISA, as described above.

## Quantitative real-time PCR

VK2/E6E7 cells from the co-culture were harvested for total RNA extraction and quantitative real-time PCR. Total RNA from VK2/E6E7 cells was isolated using TRIzol reagent (Invitrogen, San Diego, USA) according to the manufacturer's instructions. cDNA was synthesized using the PrimeScript RT Reagent Kit (TaKaRa Bio Inc., Japan) on a GeneAmp PCR System 9700 thermal cycler (Applied Biosystems). Quantitative real-time PCR was performed on an Applied Biosystems 7500 Real-Time PCR System using Hieff qPCR SYBR Green Master Mix (11204ES08, Yeasen). All reactions were run in triplicate in three independent experiments. Expression levels of *TLR2*, *TLR4*, and *MyD88* were normalized to *GAPDH* and calculated relative to the control group using the $2^{-\Delta\Delta Ct}$ method.

## Western blot assay

Total protein from VK2/E6E7 cells was extracted using ice-cold RIPA lysis buffer (50 mM Tris-HCl, pH 7.4; 150 mM NaCl; 1% NP-40; 0.25% sodium deoxycholate; 1 mM EDTA) supplemented with protease and phosphatase inhibitors (1 µg/mL aprotinin, pepstatin, and leupeptin; 1 mM $Na_3VO_4$; 1 mM NaF; 1 mM PMSF). Lysates were collected by centrifugation, and protein concentrations were determined using an Enhanced BCA Protein Assay Kit (Beyotime, China). Protein solutions (30 µg per lane) were separated on 10% SDS–PAGE gels and transferred onto polyvinylidene difluoride (PVDF) membranes (Bio-Rad, Hercules, CA, USA). Membranes were blocked with 5% non-fat milk in TBST for 1 h at room temperature and then incubated overnight at 4°C with primary antibodies against TLR2, TLR4, MyD88, phospho-NF-κB p65 (p-p65), and phospho-IκBα (all from Santa Cruz Biotechnology, USA). After washing, membranes were incubated with HRP-conjugated goat anti-rabbit IgG secondary antibodies (Santa Cruz Biotechnology, USA) for 1 h at room temperature. Protein bands were visualized using an enhanced chemiluminescence (ECL) detection kit (Thermo Fisher Scientific, USA), and band intensities were quantified by densitometric analysis using Quantity One software (Bio-Rad, USA).

## Murine model of vaginal candidiasis

The murine model of vulvovaginal candidiasis has been extensively described in prior studies (24, 25). Female C57BL/6 mice aged between 6 and 8 weeks were procured from the Animal Laboratory of Zhejiang University and maintained in isolator cages on ventilated racks. The mice were randomly divided into three groups, with 6 mice allocated to each group. The animals were administered subcutaneously with 0.1 mg of estrogen (β-estradiol 17-valerate; Sigma, USA) dissolved in 0.1 mL of sesame oil 72 h prior to the infection.

*C. albicans* cells at the stationary phase were harvested, washed three times with sterile, endotoxin-free PBS, and resuspended. Estrogen-treated mice were anesthetized with isoflurane, and 20 µL of the standardized suspension (containing $5 \times 10^6$ yeast cells) was gently instilled into the vaginal canal. Naïve control mice received 20 µL of sterile PBS alone.

## Vaginal lavage collection and fungal burden quantification

At day 3 post-inoculation, vaginal lavage fluid was collected using sterile PBS. Lavage samples were serially 10-fold diluted and plated on YPD agar for CFU enumeration. Plates were incubated until colonies were countable, and fungal burden was calculated as CFU per milliliter (CFU/mL) and $\log_{10}$-transformed for analysis.

## Vaginal lavage

Mice were euthanized 3 days post-infection, and vaginal lavage was collected by gently flushing the vaginal canal with 100 µL PBS containing protease inhibitors (cOmplete, EDTA-free; Roche). An aliquot from each sample was used to determine fungal burden and to enumerate polymorphonuclear leukocytes (PMNs). The remaining lavage fluid was centrifuged at 3,500 rpm for 5 min at 4°C to remove cellular debris, and the supernatant was stored at −80°C for subsequent analyses.

## Polymorphonuclear leukocytes (PMNs) quantification

Vaginal lavage samples (10 µL per animal) were applied to Tissue-Tek Superfrost Plus Gold slides (Thermo Fisher Scientific, USA), air-dried, and fixed with CytoPrep spray fixative (Thermo Fisher Scientific, USA) at room temperature. Slides were then stained using the Papanicolaou (Pap) method. PMNs were identified based on characteristic staining, morphology, and the presence of multilobed (typically trilobed) nuclei. Each smear was examined using a light microscope. PMNs were manually counted in five non-overlapping fields per smear.

## Histopathological analysis and PMN counting

Vaginal tissues were collected for histopathological analysis. Sections 5 µm thick were prepared and mounted on glass slides. Neutrophils were detected by immunohisto-chemistry using a mouse myeloperoxidase (MPO) detection kit (Yaji Biological, China). Briefly, deparaffinized and rehydrated sections were subjected to antigen retrieval as recommended by the manufacturer, followed by incubation with primary anti-MPO antibody (20 µL per section) overnight at 4°C. After equilibration to room temperature, sections were incubated with secondary antibody (50 µL) and then with streptavidin-biotin-peroxidase complex (SABC, 50 µL). Immunoreactivity was assessed with 3,3′-di-aminobenzidine (DAB) as the chromogen. Histopathological changes and MPO-positive PMNs were examined under a light microscope (Olympus, Japan). For quantification, neutrophils were counted in five randomly selected, non-adjacent fields per section at a defined magnification, and the mean PMN count per section was calculated for each animal.

## Statistical analysis

Data are presented as mean ± standard error of the mean (SEM) or mean ± standard deviation (SD) unless otherwise indicated. For comparisons between two groups, an unpaired two-tailed Student's $t$-test was used. For comparisons involving multiple groups and/or factors, two-way analysis of variance (ANOVA) was performed where appropriate. $P$ value < 0.05 was considered statistically significant. Statistical analyses and graphing were performed using GraphPad Prism 8.0.2 (GraphPad Software, USA).

## RESULTS

### *lat1* deletion negatively affects *Candida* proliferation and yeast-to-hyphae morphogenesis

To investigate the impact of *lat1* on *C. albicans* growth and morphogenesis, we constructed a *lat1*-null (*lat1Δ/Δ*) mutant using *C. albicans* SC5314 and CRISPR-Cas9. To assess the growth defect, WT and *lat1Δ/Δ* mutant strains were grown in YPD at 30°C, and growth was compared by measuring $OD_{600}$ at 1, 3, 6, 12, and 24 h. The *lat1Δ/Δ* mutant displayed lower $OD_{600}$ relative to that of the WT, particularly at later time points (**$P$ < 0.01, ***$P$ < 0.001, and ****$P$ < 0.0001, Fig. 1B), suggesting impaired growth.

We also compared yeast-hyphae morphogenesis of WT and *lat1Δ/Δ* mutant under hypha-inducing conditions (37°C, in the presence of 10% serum). WT *Candida* cells readily grew into germ tubes by 1 h, and most cells elongated into true hyphae by 3–6 h

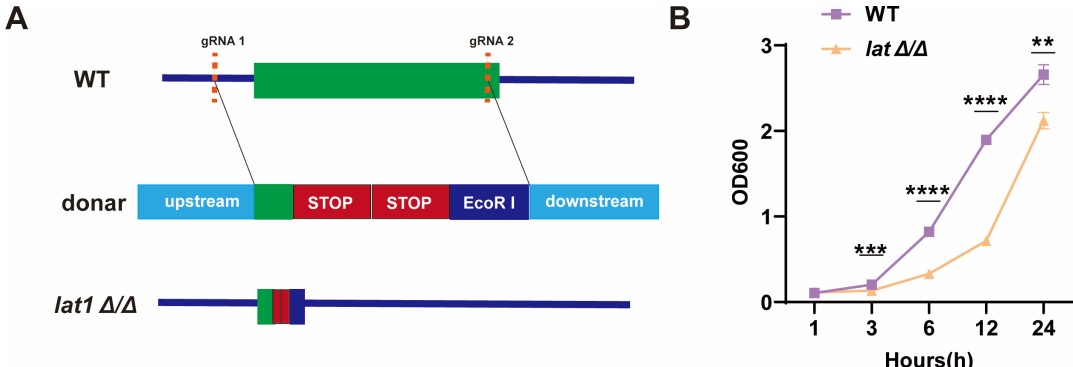

**FIG 1** Generation of a *lat1Δ/Δ* mutant reveals impaired *in vitro* growth of *C. albicans*. (A) Schematic of CRISPR-Cas9-mediated inactivation of *lat1*. sgRNAs targeting the *lat1* coding region were designed, and a repair oligonucleotide donor was used to introduce two in-frame stop codons (TAA and TAG) into the LAT1 open reading frame. Specifically, the endogenous sequence TTTGCC was replaced with TAATAG, and an EcoRI restriction site was introduced to facilitate screening. (B) Growth kinetics of WT and the lat1Δ/Δ mutant were monitored by measuring optical density at 600 nm ($OD_{600}$) at the indicated time points (1, 3, 6, 12, and 24 h). Data are presented as mean ± SEM from three independent experiments. Statistical significance was determined by unpaired *t*-test (**$P< 0.01$; ***$P < 0.001$; ****$P< 0.0001$).

(Fig. 2B). In contrast, *lat1Δ/Δ* mutant cells remained predominantly as yeast cells throughout the 6-hour time period, with very few cells presenting as short germ tubes (Fig. 2B). Quantitative comparison confirmed that the percentage of germ-tube-forming cells in *lat1Δ/Δ* mutant cultures was significantly lower than that in WT at all time points (****$P < 0.0001$; Fig. 2A).

The *in vitro* growth defect mediated by the *lat1Δ/Δ* mutant was mirrored by the lower fungal burden found in the mouse model of vulvovaginal candidiasis. On day 3 after intravaginal inoculation, colony-forming unit (CFU) counts from vaginal lavage fluid were substantially higher in mice infected with WT strain than those of the *lat1Δ/Δ* mutant-infected mice (****$P < 0.0001$; Fig. 3). Together, these results indicate that *lat1* contributes to *C. albicans* fitness and yeast-hyphae morphogenesis.

## *lat1* is associated with *C. albicans* adhesion to vaginal epithelial cells

Adhesion of *C. albicans* cells to vaginal epithelial cells is essential for the formation of biotic biofilms, fungal invasion of host tissues, and the pathogenesis of recurrent vulvovaginal candidiasis. To determine the contribution of *lat1* to fungal adherence, we performed an immunofluorescence-based adhesion assay using VK2/E6E7 epithelial cells. VK2/E6E7 cells were visualized with phalloidin staining (red), and *C. albicans* cells were detected using an anti-*C. albicans* antibody (green) (Fig. 4B). Adhesion was

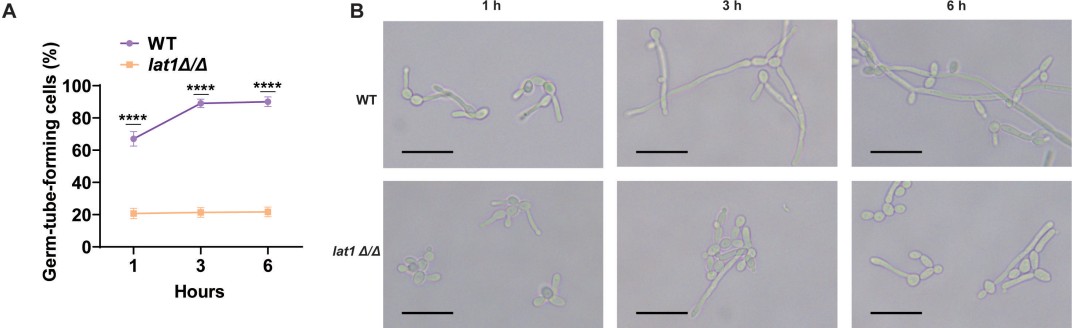

**FIG 2** Loss of *lat1* impairs hyphal morphogenesis in *C. albicans*. (A) Germ tube formation was significantly reduced in the *lat1Δ/Δ* mutant compared with WT at all time points examined under standard hypha-inducing conditions at 37°C (1, 3, and 6 h; ****$P < 0.0001$). (B) Representative bright-field images and corresponding quantification over a 6-hour time course show that WT cells formed germ tubes by 1 h and developed elongated filaments by 3–6 h, whereas *lat1Δ/Δ* cells remained predominantly in the yeast form with only occasional short germ tubes. Scale bars: 100 µm.

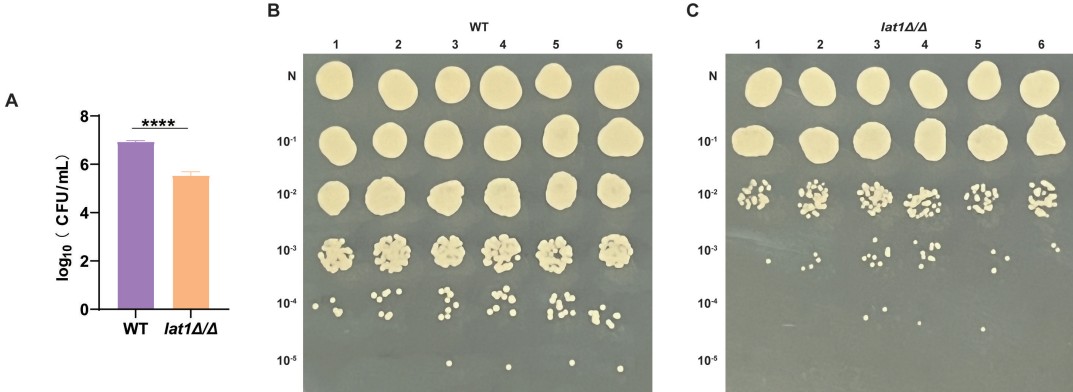

**FIG 3** *lat1* deletion reduces vaginal fungal burden in a murine model of vulvovaginal candidiasis. (A) Vaginal fungal burden at day 3 post-inoculation, expressed as $\log_{10}$(CFU/mL) recovered from vaginal lavage fluid from mice infected intravaginally with WT or the *lat1Δ/Δ* mutant. Data are presented as mean ± SEM (*n* = 6 mice per group). *P* values were determined using an unpaired *t*-test (****$P$ < 0.0001). (B–C) Representative serial dilution spot assays of vaginal lavage samples from individual mice infected with WT (B) or *lat1Δ/Δ* (C). *N* indicates undiluted lavage, followed by 10-fold serial dilutions as indicated.

quantified as the number of fungal cells attached per epithelial cell, scored from five randomly selected microscopic fields per replicate. WT *C. albicans* adhered robustly to VK2/E6E7 cells, whereas the *lat1Δ/Δ* mutant showed markedly reduced binding (****$P$ < 0.0001). Together, these results indicate that *lat1* is required for efficient adhesion of *C. albicans* to vaginal epithelial cells.

## *lat1* deletion attenuates TLR2/4-MyD88-NF-κB signaling in vaginal epithelial cells

Given the impaired proliferation, morphogenesis, and adhesion of the *lat1Δ/Δ* mutant, we next asked whether deletion of *lat1* also affects host innate immune signaling in vaginal epithelial cells. It is known that in *C. albicans* infection, Toll-like receptors (TLRs) are key sensors of invading fungi and are involved in initiating host immune defense

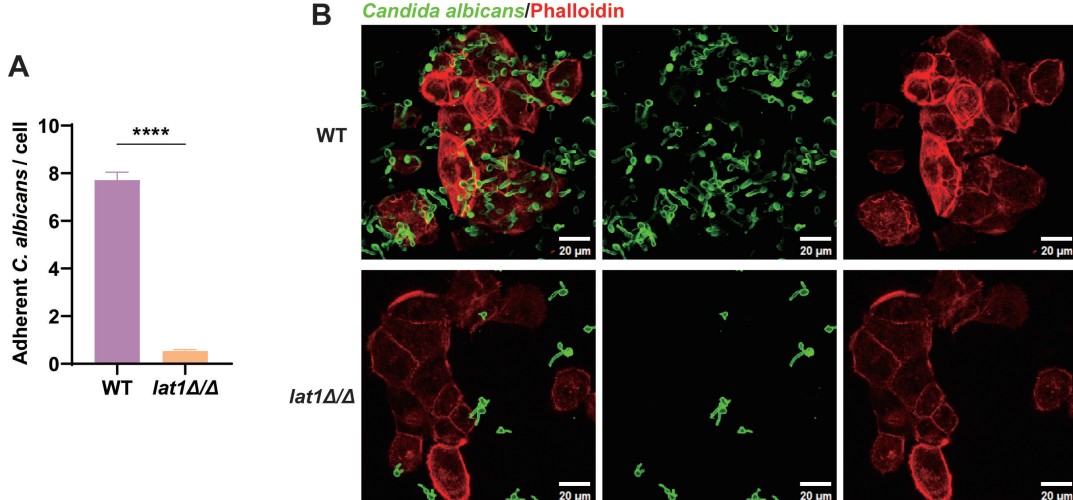

**FIG 4** lat1 deletion impairs adhesion of *C. albicans* to VK2/E6E7 vaginal epithelial cells. (A) Quantification of adherent *C. albicans* per epithelial cell following co-culture of VK2/E6E7 cells with WT or the *lat1Δ/Δ* mutant. Adhesion was scored as the number of fungal cells attached per epithelial cell from five randomly selected microscopic fields per replicate. WT showed robust adherence (mean, 7.741 fungi/cell), whereas the *lat1Δ/Δ* mutant exhibited markedly reduced binding (mean, 0.5382 fungi/cell). Data are presented as mean ± SEM from three independent experiments. *P* values were determined using an unpaired *t*-test (****$P$ < 0.0001). (B) Representative confocal images of VK2/E6E7 cells co-cultured with WT or *lat1Δ/Δ*. Epithelial cells were stained with phalloidin (red), and *C. albicans* cells were labeled with an anti-*C. albicans* antibody (green). Scale bar: 20 µm.

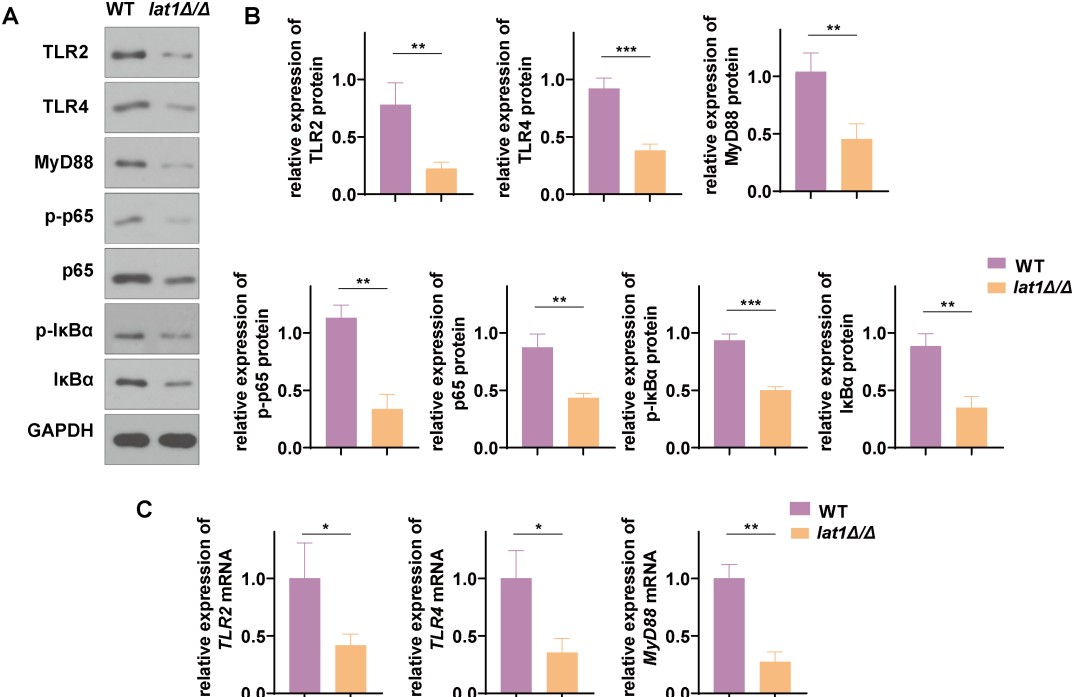

**FIG 5** *lat1* deletion attenuates TLR2/4–MyD88–NF-κB signaling in VK2/E6E7 vaginal epithelial cells. (A) Representative immunoblots showing TLR2, TLR4, MyD88, phospho-NF-κB p65 (P-P65), total p65, phospho-IκBα (p-IκBα), total IκBα, and GAPDH. (B) Densitometric quantification of protein levels normalized to GAPDH and expressed relative to the WT group. (C) Relative mRNA levels of TLR2, TLR4, and MyD88 determined by RT-qPCR, normalized to GAPDH and expressed relative to the WT group. Data are presented as mean ± SEM from three independent experiments. *P* values were determined using an unpaired *t*-test; *$P < 0.05$, **$P < 0.01$, and ***$P < 0.001$.

mechanisms (26). In mammals, most TLRs signal via the adaptor protein MyD88 to activate NF-κB (27). To determine whether *lat1* is associated with TLR2/4-MyD88-NF-κB signaling, VK2/E6E7 cells were co-cultured with WT or *lat1Δ/Δ* mutant strains, respectively, for 24 h. Relative mRNA levels of *TLR2*, *TLR4*, and *MyD88* were measured by qPCR, and protein syntheses of TLR2, TLR4, MyD88, phospho-p65 (p-p65), total p65, phospho-IκBα (p-IκBα), and total IκBα were assessed by western blotting. Compared with the WT strain, the *lat1Δ/Δ* mutant strain induced significantly lower transcript levels of *TLR2*, *TLR4*, and *MyD88* in VK2/E6E7 cells (*$P < 0.05$, **$P < 0.01$; Fig. 5C). Consistently, protein syntheses of TLR2, TLR4, MyD88, p-p65, and p-IκBα were also reduced in *lat1Δ/Δ*-infected epithelial cells (**$P < 0.01$, ***$P < 0.001$; Fig. 5A and B). These results suggest that *lat1* contributes to robust activation of the TLR2/4-MyD88-NF-κB pathway, and infection with the *lat1Δ/Δ* mutant only triggers weak epithelial innate immune signaling.

## Inflammatory cytokine production is attenuated by *lat1* deletion

To further confirm whether *lat1* affects the production of inflammatory cytokines in vaginal epithelial cells, we measured TNF-α, IL-6, IL-1β, IL-10, and CCL2 in the supernatants of VK2/E6E7 cells co-cultured with WT or *lat1Δ/Δ* mutant strains for 24 h. ELISA showed that VK2/E6E7 cells co-cultured with *lat1Δ/Δ* mutant produced markedly lower levels of inflammatory factors compared to cells exposed to the WT strain (*$P < 0.05$, **$P < 0.01$, and ***$P < 0.001$; Fig. 6A)

To assess whether *lat1* deletion also modulates cytokine responses *in vivo*, we measured TNF-α, IL-6, IL-1β, IL-10, and CCL2 in diluted vaginal lavage fluid from mice with vulvovaginal candidiasis using ELISA. Exposure to WT or *lat1Δ/Δ* mutant strains generally resulted in increased cytokine production relative to that of uninfected controls, whereas cytokine levels in *lat1Δ/Δ*-infected mice were consistently lower than those of WT-infected mice (*$P < 0.05$ for the indicated comparisons; Fig. 6B). Taken together, these *in*

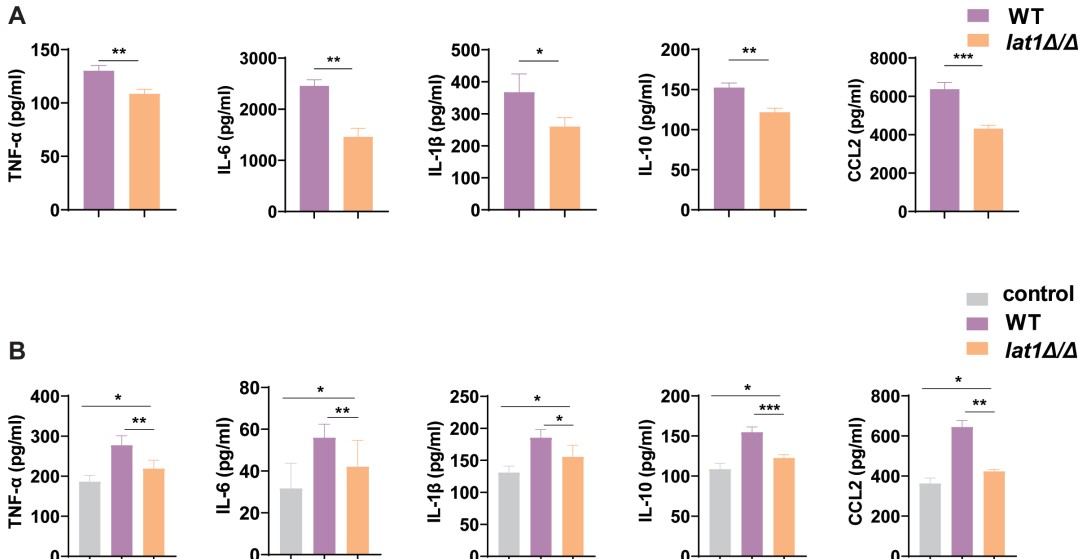

**FIG 6** *lat1* deletion attenuates inflammatory cytokine production *in vitro* and *in vivo*. (A) Cytokine levels (TNF-α, IL-6, IL-1β, IL-10, and CCL2) in culture supernatants of VK2/E6E7 cells co-cultured for 24 h with WT or the *lat1Δ/Δ C. albicans* strain, quantified by ELISA. Data are presented as mean ± SD (*n* = 3). *P* values were determined using an unpaired *t*-test. (B) Cytokine levels in vaginal lavage fluid from uninfected control mice and mice infected with WT or *lat1Δ/Δ*, quantified by ELISA. Data are presented as mean ± SD (*n* = 5). *P* values were determined by one-way ANOVA with Tukey's multiple comparisons test. Significance is indicated as *$P$ < 0.05, **$P$ < 0.01, and ***$P$ < 0.001.

*vitro* and *in vivo* data support the conclusion that *lat1* deletion results in attenuated inflammatory cytokine responses in vulvovaginal candidiasis.

## *lat1* deletion reduces neutrophil recruitment to the site of vaginal infection

Very few PMNs were observed in the Papanicolaou-stained smears of vaginal lavage from uninfected mice. Infection with WT *C. albicans* led to abundant PMNs in the lavage smears, whereas exposure to *lat1Δ/Δ* mutant induced fewer PMNs (Fig. 7B). Quantification of PMNs per smear showed that *lat1Δ/Δ*-infected mice had 38.4 ± 3.78 PMNs per smear, compared with 92.4 ± 4.83 PMNs per smear in WT-infected mice (***$P$ < 0.0005, ****$P$ < 0.0001; Fig. 7A), indicating that WT strain elicits stronger local vaginal inflammation than the *lat1Δ/Δ* mutant strain.

Histopathological analysis of vaginal tissue 72 h post-infection showed that, in comparison to WT strain, *lat1Δ/Δ* strain induced visibly fewer MPO-positive cells in the vaginal lamina propria. Quantification of neutrophils in five randomly selected, non-adjacent fields per section (20× objective) confirmed that neutrophil numbers were significantly lower in *lat1Δ/Δ*-infected mice than in those infected with the WT strain (*$P$ < 0.05; Fig. 8A and B).

Together with the attenuated cytokine responses, these findings suggest that *lat1* contributes to efficient recruitment of neutrophils to the infected vaginal mucosa.

## DISCUSSION

Genetic factors of *C. albicans* that determine its pathogenicity in vulvovaginal candidiasis are complex and not fully understood. In this study, we report for the first time the crucial role of a specific amino acid transporter, L-type amino acid transporter 1 (Lat1) in *C. albicans*, in the pathogenesis of VVC. By constructing a *lat1Δ/Δ* mutant strain and comparing it to the wild-type parent strain, we demonstrated that the absence of *lat1* leads to significantly impaired cell proliferation, yeast-hyphae morphogenesis, and adhesion of *C. albicans* to the vaginal epithelium. Furthermore, the *lat1Δ/Δ* strain induced a much weaker innate immune response in host cells and tissues, with reduced

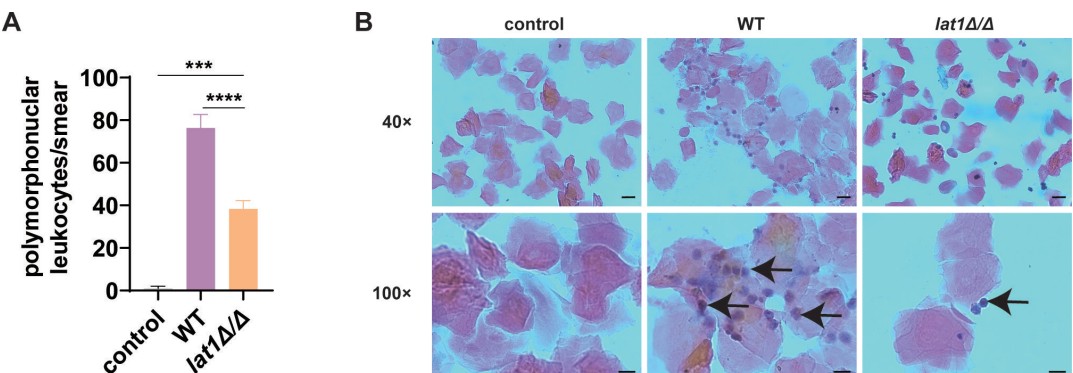

**FIG 7** *lat1* deletion reduces neutrophil recruitment to the vaginal lumen during infection. Vaginal lavage was collected at 72 h post-infection and processed for Papanicolaou (Pap) staining. (A) Quantification of polymorphonuclear leukocytes (PMNs) in vaginal lavage smears from uninfected controls and mice infected with WT or the *lat1Δ/Δ* mutant. PMNs were identified by characteristic morphology and multilobed nuclei and counted in five non-overlapping fields per smear; the mean value per animal is shown. WT-infected mice showed higher PMN counts (92.4 ± 4.83 PMNs/smear) than *lat1Δ/Δ*-infected mice (38.4 ± 3.78 PMNs/smear) (***$P < 0.0005$, ****$P < 0.0001$). Data are presented as mean ± SD ($n = 5$). $P$ values were determined by one-way ANOVA with Tukey's multiple comparisons test. (B) Representative Pap-stained smears; arrows indicate PMNs.

secretion of inflammatory cytokines and fewer immune cells (neutrophils) recruited to the infection site. In summary, our research highlights *lat1* as a key contributor to the virulence and pathogenicity of *C. albicans* in VVC.

Lat1 is an important amino acid transporter that facilitates the transmembrane transport of specific amino acids and plays a crucial role in various metabolic pathways (28–31). Lat1 may serve as a critical regulator for the influx and efflux of essential amino acids (EAA) in cells (32), mediating the entry of neutral EAA, such as glutamine, leucine, and isoleucine, into cells while exchanging them for intracellular substrates and regulating various cell growth and metabolic processes (30). Our interest in *lat1* stemmed from a previous transcriptomic analysis suggesting that *lat1* might be central to *C. albicans*' interaction with host cells. We hypothesized that Lat1-mediated amino acid transport could be important for fungal growth in nutrient-limited or stressful host niches, thereby influencing *Candida* virulence. Supporting this hypothesis, we observed that deleting *lat1* resulted in a slower growth of *C. albicans* in rich medium. Similarly, in the mouse vaginal tract, the fungal burden of *lat1Δ/Δ* was significantly lower than that of WT. This suggests that *lat1* contributes to optimal fungal proliferation, possibly by ensuring sufficient uptake of amino acids needed for protein synthesis and growth.

*C. albicans* is known to be able to colonize various mucosal surfaces as a commensal organism, including the epidermis, oral cavity, gastrointestinal tract, and urogenital tract, without evident clinical symptoms. Under certain predisposing factors, *C. albicans* can transition from the symbiotic state into a pathogen (31). This transition involves multiple cellular responses such as proliferation, yeast-to-hyphae morphological transition, adhesion to and invasion of the vaginal epithelial cells (33), immunomodulation of host responses, and the release of inflammatory factors. These processes require the synthesis, transport, and utilization of various amino acids, making amino acid transporters indispensable. The *lat1Δ/Δ* strain constructed in our study showed a significantly lower capacity in these important cellular processes than the WT strain, indicating a substantial impact of *lat1* on the pathogenicity of *C. albicans* and its interactions with host cells in VVC.

It is known that Toll-like receptors (TLRs) play a crucial role in sensing invading fungal pathogens and initiating host immune defense mechanisms (26) during *C. albicans* infections. In mammals, most TLRs activate NF-κB through the adaptor protein MyD88 (27). Whether *lat1* affects the development and severity of local vaginal inflammation by influencing the TLRs-MyD88-NF-κB signaling pathway and its downstream inflammatory factors remains unknown. In this study, we used a co-culture model of *C. albicans*

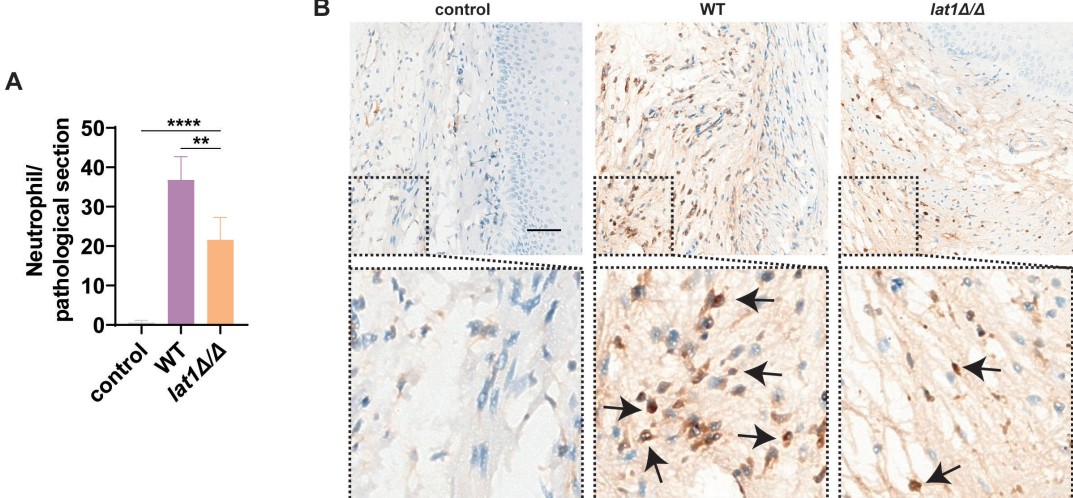

**FIG 8** *lat1* deletion reduces MPO-positive neutrophil infiltration in vaginal tissue. Vaginal tissues were collected at 72 h post-infection and analyzed by myeloperoxidase (MPO) immunohistochemistry. (A) Representative images showing MPO-positive cells in the vaginal lamina propria from uninfected controls and mice infected with WT or *lat1Δ/Δ*. (B) Quantification of MPO-positive neutrophils counted in five randomly selected, non-adjacent fields per section at 20× magnification. Data are presented as mean ± SD (*n* = 5). *P* values were determined by one-way ANOVA with Tukey's multiple comparisons test (**$P < 0.01$, ****$P <0.0001$).

and vaginal epithelial cells and found *lat1Δ/Δ* led to a reduction in TLR2 and TLR4 production in vaginal epithelial cells, along with significant decreases in MyD88 and its downstream p-IκBα and NF-κB p-p65. Additionally, inflammatory factors, such as tumor necrosis factor-α (TNF-α), IL-6, IL-1β, IL-10, and chemokine CC ligand-2 (CCL2), were also significantly reduced. These experimental results, supported by our *in vivo* findings, suggest that the *lat1* in *C. albicans* may influence vaginal epithelial inflammation via the TLR2/4/MyD88/NF-κB signaling pathway. Inactivation or deletion of *lat1* in *C. albicans* may impact its recognition of host surface TLRs, resulting in a weakened inflammatory response, reduced secretion of inflammatory factors (34), decreased production of chemokines, and impaired recruitment of immune cells such as neutrophils to the infection site. Additionally, the phagocytic activity of macrophages, nitric oxide production, and myeloperoxidase activity may also be affected, and these aspects require further in-depth research.

Besides the TLRs-MyD88-NF-κB signaling pathway, associations between *lat1* and inflammation mediated by other pathways cannot be excluded, such as the mTOR pathway (1, 35), TRIF pathway, and other non-NF-κB pathways, including the MAPK pathway and AP-1 pathway (36). Further investigation is needed.

This study has several limitations that should be noted. First, we used the *C. albicans* laboratory reference strain SC5314 which originated from an oral candidiasis patient, as the WT strain. While SC5314 is a well-characterized and widely used reference strain, it may not fully represent clinical isolates from the vagina in terms of virulence traits(37). We acknowledge our results should ideally be validated with clinical isolates from patients with confirmed VVC. Second, we did not construct a *lat1* complementation strain to further confirm that the observed phenotypic defects were due to the loss of *lat1*. We attempted to minimize the chance of off-target effects by sequencing the edited locus and using a precise CRISPR edit rather than a random insertion, and the observed phenotypes are consistent with an expected metabolic handicap. Third, our adhesion assay was limited to a 2-hour co-incubation. A 24-hour biofilm assay could provide additional insights into how *lat1* affects the ability of *C. albicans* to form stable, mature communities on vaginal cells. Finally, our *in vivo* analysis was conducted at a single time point (3 days post-infection). It is possible that the *lat1Δ/Δ* strain, although slower to establish an infection, might eventually cause inflammation if more time was given. It

remains possible that, at later time points, the mutant could "catch up" to the wild type in terms of host inflammatory responses. We did not extend the infection beyond 3 days because untreated murine vaginitis typically self-resolves within approximately one week. It is possible that the difference between WT and *lat1Δ/Δ* mutant strains might lessen if the post-infection observational time was extended.

In summary, we have demonstrated that *C. albicans* with defective *lat1* exhibit reduced virulence in VVC, as evidenced by impaired growth, morphogenesis, adhesion, and host immune activation. These results highlight the importance of amino acid transport in fungal pathogenicity and suggest that metabolic factors such as *lat1* can profoundly influence the host-pathogen interactions. Further exploration of the role of *lat1* in inflammation caused by *C. albicans* is warranted, which may provide novel approaches for the clinical management of inflammation associated with *C. albicans* infections.

## Conclusion

*lat1* plays an important role in the pathogenicity of *C. albicans*, influencing cell proliferation, morphogenesis, adherence to host cells, and its ability to elicit local inflammatory responses through the TLR2/4-MyD88-NF-κB pathway in the vaginal epithelium. This study is the first one to establish a link between *lat1* and *C. albicans* virulence, underscoring the significant role of amino acid transport in the pathogenesis of fungal infections. Understanding how amino acid transporters impact *C. albicans* infections provides a new perspective that could inform the development of novel strategies for the prevention and treatment of candidiasis and related microbial infectious diseases.

## ACKNOWLEDGMENTS

We thank Qianbing Zhao from the Core Facilities, Zhejiang University School of Medicine, for their technical support.

This research was supported by the Joint Funds of Zhejiang Provincial Natural Science Foundation of China under Grant No. LHDMZ25H040001, the Zhejiang Provincial Natural Science Foundation of China under Grant No. LY18H040003, the National Natural Science Foundation of China under Grant No. 82571881, No. 81801405.

Y.L. and C.W. contributed equally to this work. Y.L., P.D., and J.J. performed the experiments and analyzed the data. Y.L., C.W., C.L., K.C., T.Z., A.L., P.C., W.X. and T.H. assisted with data interpretation. Y.L., C.W. drafted the initial manuscript. J.Z., W.L., L.M., and Y.H. supervised the study. C.W., L.M., and Y.H. revised the manuscript. Y.H. and L.M. are both corresponding authors. All authors read and approved the final manuscript.

## AUTHOR AFFILIATIONS

[1]Department of Gynecology, Women's Hospital, Zhejiang University School of Medicine, Hangzhou, China
[2]School of Basic Medical Sciences and Forensic Medicine, Hangzhou Medical College, Hangzhou, Zhejiang, China
[3]Zhejiang Maternal Child and Reproductive Health center, Hangzhou, China
[4]Institute of Immunology and The Second Affiliated Hospital, Zhejiang University, Hangzhou, China

## AUTHOR ORCIDs

Cheng Wu http://orcid.org/0009-0007-0073-2905
Linjuan Ma http://orcid.org/0009-0001-8868-793X
Yizhou Huang http://orcid.org/0000-0003-4456-6543

## FUNDING

This research was supported by the Joint Funds of Zhejiang Provincial Natural Science Foundation of China under Grant No. LHDMZ25H040001, the Zhejiang Provincial Natural Science Foundation of China under Grant No. LY18H040003, the National Natural Science Foundation of China under Grant No. 82571881, No. 81801405.

## AUTHOR CONTRIBUTIONS

Yibing Lan, Conceptualization, Data curation, Methodology, Visualization, Writing – original draft, Writing – review and editing | Cheng Wu, Writing – original draft, Writing – review and editing | Peng Du, Conceptualization, Formal analysis, Project administration | Jie Jiao, Data curation, Formal analysis, Methodology | Chunming Li, Data curation, Formal analysis, Investigation, Methodology, Software | Ketan Chu, Data curation, Formal analysis, Investigation, Methodology, Software, Validation | Tao Zhang, Conceptualization, Data curation, Investigation, Methodology, Validation | Peiqiong Chen, Conceptualization, Data curation, Investigation, Methodology, Resources | An Li, Data curation, Formal analysis, Investigation, Methodology | Wenxian Xu, Conceptualization, Data curation, Methodology, Resources, Supervision | Xinyi Ying, Conceptualization, Data curation, Formal analysis, Investigation, Methodology, Supervision, Validation, Visualization | Jianhong Zhou, Data curation, Formal analysis, Funding acquisition, Investigation, Methodology, Resources, Supervision, Writing – original draft | Wenlong Lin, Conceptualization, Formal analysis, Funding acquisition, Investigation, Methodology, Project administration, Resources, Supervision, Validation, Visualization | Linjuan Ma, Funding acquisition, Investigation, Methodology, Resources, Supervision, Writing – original draft | Yizhou Huang, Conceptualization, Formal analysis, Funding acquisition, Methodology, Project administration, Resources, Supervision, Visualization

## DATA AVAILABILITY

The data sets used and/or analyzed during the current study are available from the corresponding author on reasonable request.

## ETHICS APPROVAL

This study was conducted in accordance with the ethical standards of the Laboratory Animal Welfare & Ethics Committee of Women's Hospital, School of Medicine, Zhejiang University. The experimental protocols involving animals were reviewed and approved by the committee (approval number: AE 20230006). All procedures were carried out in compliance with relevant guidelines and regulations to ensure the welfare of the animals.

## ADDITIONAL FILES

The following material is available online.

### Open Peer Review

**PEER REVIEW HISTORY (review-history.pdf).** An accounting of the reviewer comments and feedback.

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
