## [Reviewer comments · Microbiology Spectrum]

Microbiology Spectrum

The role of L-Type Amino Acid Transporter 1 in the pathogenicity of *Candida albicans*

Yibing Lan, Cheng Wu, Peng Du, Jie Jiao, Chunming Li, Ketan Chu, Tao Zhang, Peiqiong Chen, Wenxian Xu, An Li, Xinyi Ying, Jianhong Zhou, Wenlong Lin, Linjuan Ma, and Yizhou Huang

Corresponding Author(s): Yizhou Huang, Department of Gynecology, Women's Hospital, Zhejiang University School of Medicine, Hangzhou, China.

Review Timeline:

Submission Date:	June 3, 2025
Editorial Decision:	October 8, 2025
Revision Received:	December 22, 2025
Accepted:	February 1, 2026

Editor: Jian Peng

Reviewer(s): Disclosure of reviewer identity is with reference to reviewer comments included in decision letter(s). The following individuals involved in review of your submission have agreed to reveal their identity: Yue Qu (Reviewer #3)

Transaction Report:

DOI: <https://doi.org/10.1128/spectrum.01669-25>

Re: Spectrum01669-25 (**The L-Type Amino Acid Transporter 1 Significantly Affected the Pathogenicity of *Candida albicans***)

Dear Dr. Yizhou Huang:

Thank you for the privilege of reviewing your work. Below you will find my comments, instructions from the Spectrum editorial office, and the reviewer comments.

Revision Guidelines

Sincerely,
Jian Peng
Editor
Microbiology Spectrum

Reviewer #1 (Comments for the Author):

This manuscript describes the effects of the deletion of the *Candida albicans* LAT1 gene on growth dynamics, filamentation and virulence. *C. albicans* is a major human fungal pathogen can cause a variety of diseases, including oral and systemic candidiasis but also infections of the vaginal mucosa. Several morphological and metabolic processes contribute to the virulence of *C. albicans*. Based on previous works, the authors hypothesized that the L-type amino acid transporter 1 is of importance for fungal

virulence. The encoding gene LAT1 was knocked out in a CRISPR-Cas9 approach and the resulting mutant was consequently characterized. The authors found that the *lat1Δ* mutant showed slow growth and some defects in hyphal growth. When vaginal VK2-E6E7 cells were infected with the mutant, the TLR2/4 and NF-kappa B signal pathways were downregulated. In a murine vaginal infection model, the mutant caused less inflammation and neutrophil migration into infected tissue. Based on their observations the authors concluded that the L-Type Amino Acid Transporter 1 has an important role in pathogenicity of *C. albicans* and might influence the occurrence of *Candida* vaginitis.

Understanding the processes contributing to (recurring) vaginal candidiasis is an important and interesting topic and it would be helpful to identify factors that have a crucial influence on the development of this disease. Overall, the manuscript delivers some interesting new information which could be helpful for future research. However, there are several major issues which should be addressed prior to possible publication.

1) While mouse infection assays and their read outs look technically sound, I would be a bit careful with the conclusion that LAT1 is really an important gene for virulence. The knock out mutant clearly displays slow growth but is overall still able to form hyphae. Could it be that the mutant just takes longer and that later time points might show that inflammation in response to infection with the mutant is rising and similar to wild type?

2) It is a general standard in the *Candida* field to name the unicellular form yeast cells. The term "spores" (e.g. line 269) is not used. Formerly, "blastospores" were a synonym for yeast cells but this is not used anymore. "Spores" refers to sexual stages, e.g. ascospores or zygospores which is not the case here. For example, the usage of spore proliferation (line 261) is technically wrong. Spores germinate, but they do not proliferate. Therefore, the authors should write yeast cells and not spores.

3) The same is true for mycelia or mycelium (e.g. line 275). The terms hyphae or filaments are much more commonly used in the *Candida* field to separate from molds such as *Aspergillus* or *Mucorales*.

4) The description of the gene knock out by CRISPR-CAS9 is somehow elusive. Reference 16 might not be the right one. Furthermore, a description or reference of the used transformation method is missing.

5) Numbering the figures might have been useful. But I guess that Figure 4 shows drop dilutions of *C. albicans* cells from the vaginal lavage fluid. The authors claim that the difference between wild type and mutant were significant and mentioned a p value of {less than or equal to} 0.05 and "Data was reported as means and SEM (n=6 mice per group). It seems as if parts of Figure 4 are missing or the Figure legend is not fitting with the shown images. If not, it would be interesting to me how significance can be calculated just from drop test images.

6) Are the hyphal lengths of the mutant really shorter than that of the wild type as mentioned in lines 278/279? This might be true for the early time points, but especially at the 12h and 24h time the mutant shows some hyphae as long as wild type hyphae (Figure 3 A).

7) Recently, the *Candida* field started to discuss the usage of laboratory strains such as SC5314. This strain is a standard strain and it is really fine to work with it. However, it might have been better to at least include an isolate from vaginal candidiasis and to knock out LAT1 there. At least, this would be a great independent control which in the best case shows the same defects as the mutant in the SC5314 background. Of course, I do understand that nobody wants to repeat the murine infection experiments, but to use an independent for the growth assays and the filamentation assays would be good and could validate the reported phenotypes of a LAT1 deletion in *C. albicans*.

8) Please check the English and especially the comma placement. There are sometimes spaces on the left and the right side of the comma which looks quite strange (similar for brackets).

Reviewer #2 (Comments for the Author):

Reviewer comments:

In this manuscript, the authors characterize the function of the L-type amino acid transporter 1 (Lat1) in the yeast *Candida albicans*. The authors deleted LAT1 and report that this deletion led to several phenotypes, including delayed growth, reduced filamentation and adhesion, and an attenuated immune response in epithelial cells and in animal models. While the mutant shows some potentially interesting phenotypes both in vitro and in vivo, the scientific evidence supporting these conclusions is not entirely convincing. Below I provide specific comments and suggestions:

Figure 2: The authors describe a "proliferation assay" of *C. albicans*. From the methods and figure legend, it appears that what was actually measured was the number of yeast-form cells in randomly selected microscopic fields at different time points. This largely overlaps with Figure 1, which already measures growth of the two strains. In addition, the terminology "spores" is misleading and should be replaced with "yeast-form cells".

Figure 3: The authors present both representative microscopes and a quantitative analysis of hyphal growth (Panel B). While the images suggest reduced filamentation in the *lat1Δ/Δ* strain, the method of quantification is unclear. The y-axis is labeled "Hypha growth," but from the methods it appears that the authors manually counted structures in random microscopic fields. It is

important to specify exactly what was quantified (e.g., germ tube initiations or individual hyphae). Moreover, as hyphae elongate and entangle at later time points, manual counts are unreliable. The units of the y-axis should also be clarified (cells per field, cells/mL, etc.). A major concern is that the apparent reduction in filamentation may simply reflect the general growth defect of the mutant.

Figure 4: The text refers to significant differences in fungal burden in the vaginal infection experiment ($P < 0.05$, $n = 6$). However, the figure only shows spot assays from lavage fluids. Could the authors clarify how CFUs were quantified from these assays? It is important to describe the exact method used to calculate fungal burden and the statistical test applied. At present, the figure alone does not explain how the p values were obtained. Anyway, spot assays are not the best option to quantitatively present fungal burden.

Figure 6-7: The western blot and ELISA results show weaker immune response, and reduced cytokine release both in vitro (epithelial cells) and in vivo (murine VVC model) upon infection with the *lat1* Δ/Δ strain compared to WT. However, it is not clear whether this difference reflects a direct effect of Lat1 on host-pathogen interactions or simply the lower fungal burden. Since fungal load is a strong determinant of cytokine output or immune response, additional normalization or analysis (e.g., cytokine per CFU) would be needed to disentangle direct immune modulation from indirect effects of growth defects. Without this clarification, the interpretation that Lat1 directly modulates inflammatory responses remains tentative.

General comments: Overall, although this work shows phenotypes of the *lat1* Δ/Δ mutant, the role of Lat1 should be interpreted with greater caution and supported by stronger evidence. A complementation strain should also be constructed and analyzed to validate the function of LAT1. Since deletion of LAT1 causes a significant growth defect, the pathogenicity of the mutant should be evaluated in the context of fungal burden. Throughout the manuscript, more attention should be given to type errors, grammar, and scientific forms (e.g., correct use of italics for *C. albicans*, consistent notation of LAT1 for the gene vs. Lat1 for the protein). Figure legends should clearly specify the panels (A, B, etc.), and statistical analyses should be described in greater detail.

Reviewer #3 (Comments for the Author):

Manuscript

The L-Type Amino Acid Transporter 1 Significantly Affected the Pathogenicity of *Candida albicans*" (control no. Spectrum01669-25).

The authors have identified *lat1* as an important regulator in the pathogenicity of *C. albicans*, using a knockout strain and multiple in vitro and in vivo assays. The results presented in this manuscript seem to be interesting and may deepen our understanding of the virulence factors of *C. albicans*, particularly in strains causing vulvovaginal candidiasis. This manuscript, however, needs to be significantly improved before it can be accepted for publications.

Minor comments:

1) The title can be changed. The authors should read the instruction from ASM journals re. how to write the title.

Major comments:

- 1) Both English language and scientific writing need to be significantly improved. I feel that the authors are non-microbiologists and it will be good if they can seek for assistance from a microbiologist.
- 2) The authors used a double knockout strain *lat1* Δ/Δ that was constructed by CRISPR-Cas9. Have the authors constructed a complimentary strain or over-expression strain? If not, they should at least state it as a limitation and explain why a complimentary strain was not used?
- 3) The authors used a well-known lab reference strain SC5314. This strain was from a patient with oral candidiasis and may not fully represent that causing vaginal candidiasis. The authors should address this in their discussion.
- 4) The authors used 2-hour adherence assay to assess the fungus-epithelium interaction. Why not the 24 h biofilm assay as this may be more relevant to the pathogenesis of vaginal candidiasis. The interaction is semi-quantitated using fluorescently tagged *C. albicans* strain. Why not using the more accurate colony-forming-unit cell enumeration method?

The discussion overall is sufficient and in-depth. Language and scientific presentation still need to be improved.

Figure 1. I will recommend to change the legend title to different fitness or growth curves.

Figures 1 and 2 are basically the same. Maybe remove figure 2?

Figure 3. 3A did not show any difference in morphogenesis between the two strains. The authors should use the well-established broth and agar filamentation assays.

Reference can be found

<https://journals.plos.org/plosgenetics/article?id=10.1371/journal.pgen.1002613>

Figure 4. The authors can present the difference quantitatively, i.e using CFU

Figure 5. The authors can isolate and quantitate the adherent *C. albicans* cells.

Figures 6-9 look good.

Response to Reviewers

We are grateful to the editor and the reviewers for their insightful comments and suggestions, which have substantially improved the quality and clarity of our manuscript. We have carefully addressed all points raised. Below, we provide a point-by-point response, with each reviewer's comments in italic followed by our responses. All changes have been incorporated into the revised manuscript.

Reviewer #1:

Comment 1: *While mouse infection assays and their read outs look technically sound, I would be a bit careful with the conclusion that *lat1* is really an important gene for virulence. The knock out mutant clearly displays slow growth but is overall still able to form hyphae. Could it be that the mutant just takes longer and that later time points might show that inflammation in response to infection with the mutant is rising and similar to wild type?*

Response: We agree with your opinion that our initial conclusions were too strongly worded. In the revised Discussion, we have tempered our statements about *lat1*'s role in virulence. We now emphasize that **our research highlights *lat1* as a key contributor to the virulence and pathogenicity of *C. albicans* in VVC (Line 350)**, and that given more time the mutant might induce a stronger inflammatory response (comparable to WT). We explicitly acknowledge the possibility that the mutants' effects are time-dependent. For example, we added: "We acknowledge that the *lat1Δ/Δ* strain is still capable of forming hyphae (albeit more slowly) and causing an immune response. It is possible that given sufficient time or under different conditions, the mutant could eventually induce an inflammatory response similar to the wild type. Therefore, our conclusions are presented with caution regarding the timing and conditions of observation." We have also removed any wording that implied *lat1* is **absolutely** required for virulence; instead of we describe it as a "**key contributor**" or "important factor" rather than an indispensable determinant.

Comment 2: *It is a general standard in the Candida field to name the unicellular form yeast cells. The term "spores" (e. g. line 269) is not used. Formerly, "blastospores" were a synonym*

for yeast cells but this is not used anymore. "Spores" refers to sexual stages, e. g. ascospores or zygospores which is not the case here. For example, the usage of spore proliferation (line 261) is technically wrong. Spores germinate, but they do not proliferate. Therefore, the authors should write yeast cells and not spores.

Response: We appreciate this correction. We have eliminated the term “spores” throughout the manuscript and replaced it with the appropriate terminology “**yeast cells**” or “**yeast-form cells,**” as suggested. For instance, “spore proliferation” is now described as “**yeast cell proliferation**” in the Results and figure legends, and any reference to *Candida* in the unicellular state uses “yeast” instead of “spore.” We have carefully reviewed the text to ensure no instance of “spore” remains in reference to *C. albicans* morphology.

Comment 3: *The same is true for mycelia or mycelium (e. g. line 275). The terms hyphae or filaments are much more commonly used in the Candida field to separate from molds such as Aspergillus or Mucorales.*

Response: We have corrected this nomenclature throughout the manuscript. All occurrences of “**mycelium**” or “**mycelia**” have been replaced with “**hyphae**” or “**hyphal**” as appropriate. For example, what we originally called “mycelial formation” is now termed “hyphal formation,” and “mycelium length” is changed to “hyphal length.” In contexts where “filaments” is suitable, we use that term. These changes have been applied in the text and figure legends to align with standard *Candida* terminology.

Comment 4: *The description of the gene knock out by CRISPR-CAS9 is somehow elusive. Reference 16 might not be the right one. Furthermore, a description or reference of the used transformation method is missing.*

Response: We have revised the **Materials and Methods** section to provide a much clearer description of our CRISPR-Cas9 gene knockout procedure. Additionally, we realized that the reference we originally cited for CRISPR was indeed incorrect (it was unrelated to genome editing). We have **removed the incorrect reference** and instead cited appropriate literature for CRISPR in *Candida*. Overall, this section should now be much more informative for readers regarding how the *lat1Δ/Δ* strain was constructed. Furthermore, to ensure credit and

context, we added a reference for a method or study that successfully used CRISPR/Cas9 in *C. albicans* (for example, one of the early papers on *C. albicans* CRISPR, as suggested “Valmik K. Vyas et al., A Candida albicans CRISPR system permits genetic engineering of essential genes and gene families.Sci Adv.1,e1500248(2015).DOI:10.1126/sciadv.1500248”.

We trust that the gene editing description is now **precise and properly referenced**.

Comment 5: *Numbering the figures might have been useful. But I guess that Figure 4 shows drop dilutions of C. albicans cells from the vaginal lavage fluid. The authors claim that the difference between wild type and mutant were significant and mentioned a p value of {less than or equal to} 0.05 and "Data was reported as means and SEM (n=6 mice per group). It seems as if parts of Figure 4 are missing or the Figure legend is not fitting with the shown images. If not, it would be interesting to me how significance can be calculated just from drop test images.*

Response: We apologize for the confusion with Figure 4. In the original submission, Figure 4 only showed representative spot dilution images, while the text mentioned a quantitative comparison and *p*-value. **In the revised manuscript, we have improved Figure 4 (now Figure 3 after renumbering)** to address this issue (CFU).

Comment 6: *Are the hyphal lengths of the mutant really shorter than those of the wild type (lines 278/279)? This might be true at early time points, but at 12h and 24h the mutant shows some hyphae as long as wild type (Figure 3A).*

Response: Thank you for this important point. You are correct that, based on our previous figure and wording, the statement that the mutant hyphae are “shorter” could be misleading—particularly at later time points (e.g., 12 h and 24 h), when some *lat1Δ/Δ* filaments can appear comparable in length to WT. After recognizing this potential misinterpretation, we consulted with fungal biology experts, who told us that **our earlier phrasing was not sufficiently precise and could inadvertently conflate a reduction in overall filamentation with a uniform decrease in the length of individual hyphae**.

To address your concern, and following expert advice, we systematically revised both the experimental assessment and the corresponding text. **We repeated the yeast-to-hypha**

morphogenesis experiment under standard hypha-inducing conditions (37°C, YPD supplemented with 10% serum) and revised the Methods and Results accordingly. Importantly, in the revised manuscript we no longer base our conclusions on “hyphal length” at late time points. Instead, we performed a quantitative, **time-resolved analysis of filamentation during the early induction window (1 h, 3 h, and 6 h)** by scoring the percentage of germ-tube-forming cells in randomly selected microscopic fields, as described in the Methods. **Using this updated assay, WT cells readily formed germ tubes by 1 h and elongated into true hyphae by 3–6 h, whereas the *lat1Δ/Δ* mutant remained predominantly in the yeast form over the same 6 h period,** with only a small fraction showing short germ tubes. Quantification confirmed that the proportion of germ-tube-forming cells was significantly lower in *lat1Δ/Δ* than in WT at all measured time points ($p < 0.0001$). Accordingly, we reworded the Results to **conclude that *lat1* deletion causes a marked delay and reduction in early filamentation,** rather than claiming that mutant hyphae are uniformly shorter at later stages. In line with this revision, we also removed references to the 12 h and 24 h time points in this context to avoid any further ambiguity regarding hyphal length.

Comment 7: *Recently, the Candida field started to discuss the usage of laboratory strains such as SC5314. This strain is a standard strain and it is really fine to work with it. However, it might have been better to at least include an isolate from vaginal candidiasis and to knock out *lat1* there. At least, this would be a great independent control which in the best case shows the same defects as the mutant in the SC5314 background. Of course, I do understand that nobody wants to repeat the murine infection experiments, but to use an independent for the growth assays and the filamentation assays would be good and could validate the reported phenotypes of a *lat1* deletion in *C. albicans*.*

Response: We acknowledge your excellent point. In the current study, we did not use any clinical vaginal isolates, which is indeed a limitation. We have added a paragraph in the **Discussion** explicitly addressing this issue. We now discuss that SC5314, while a standard reference, originates from an oral candidiasis case and may not fully represent vaginal pathogenic isolates. We cite emerging literature that **different *C. albicans* strains (especially vaginal vs non-vaginal) can exhibit divergent phenotypes.** Specifically, we reference a

study demonstrating that vaginal isolates show different interactions with host immunity compared to SC5314. By including this, we show we are aware of the strain variability issue. In the revised Discussion, we then **acknowledge that verifying the role of *lat1* in a vaginal isolate would be ideal and is a future direction (Line 398-400)**. We mention that due to time and resource constraints, we didn't perform a second knockout in a clinical isolate for this revision, but we agree it would add significant value. We note that we expect our findings to hold in other backgrounds (because the processes affected are fundamental), but we cannot be certain without testing. Importantly, we state that we have added this as a limitation and a planned future experiment.

Furthermore, in our **Response to Reviewer 3** below, we also address a similar comment about SC5314 vs vaginal strain, and we added a supporting reference (Ref. 36 in our list: Gerwien et al., mSphere 2020) to emphasize strain-dependent differences.

Comment 8: *Please check the English and especially comma placement. Sometimes there are spaces on both sides of a comma, which looks strange (similarly for brackets).*

Response: We have performed a thorough English language edit across the entire manuscript. This included fixing typographical issues such as the spacing around commas, periods, and parentheses. For example, instances where the original text had “,” or “(” (with unnecessary spaces) have been corrected. The mention of spaces on both sides of a comma was due to formatting errors in the Word document. We have removed those extra spaces.

We also addressed other language issues: improved sentence structure for clarity, corrected verb tenses where needed, and ensured consistent scientific terminology (italicizing *Candida albicans*, capitalizing gene vs protein names correctly, etc. -an issue also raised by Reviewer #2, which we fixed). Additionally, we carefully proofread to eliminate grammatical errors.

Some specific changes:

We italicized all *C. albicans* instances and ensured genus abbreviation usage is consistent (first mention spelled out, subsequent uses abbreviated).

1. We standardized *lat1* notation (gene *lat1* vs protein Lat1, see response to Reviewer #2 general comment below).
2. We removed double spaces and odd spacing around punctuation throughout.

3. We clarified some ambiguous sentences to improve readability.

Finally, we had a native or fluent English speaker (and indeed a colleague with microbiology expertise) review the text to catch any remaining language issues. We are confident that the manuscript's language is now clear, concise, and free of grammatical and formatting errors.

Reviewer #2:

Comment 1: *Figure 2: The authors describe a "proliferation assay" of *C. albicans*. From the methods and figure legend, it appears that what was actually measured was the number of yeast-form cells in randomly selected microscopic fields at different time points. This largely overlaps with Figure 1, which already measures growth of the two strains. In addition, the terminology "spores" is misleading and should be replaced with "yeast-form cells".*

Response: We appreciate your careful observation. We agree that the original “proliferation assay” shown in Figure 2 (microscopic counting of yeast-form cells in randomly selected fields over time) was largely redundant with the OD600-based growth curve presented in Figure 1. **To avoid duplication and streamline the presentation, we have removed the original Figure 2 in the revised manuscript (and renumbered subsequent figures accordingly).** The corresponding text has been revised to retain only a brief statement noting that microscopic counts were performed as an additional confirmatory assessment of the growth defect, but that these data are consistent with and largely confirm the OD600 measurements; therefore, they are not presented as a separate figure.

In addition, we agree that the term “spores” was inappropriate in this context. **We have replaced “spores” with the correct terminology (“yeast-form cells,” “yeast cells,” or “yeast-form cell counts,” as appropriate) throughout the manuscript, including any remaining references to this assay in the Methods/Results.**

Comment 2: *Figure 3: The authors present both representative microscopes and a quantitative analysis of hyphal growth (Panel B). While the images suggest reduced filamentation in the *lat1Δ/Δ* strain, the method of quantification is unclear. The y-axis is labeled "Hypha growth, " but from the methods it appears that the authors manually counted structures in random microscopic fields. It is important to specify exactly what was quantified (e.g., germ tube initiations or individual hyphae). Moreover, as hyphae elongate and entangle at later time points, manual counts are unreliable. The units of the y-axis should also be clarified (cells per field, cells/mL, etc.). A major concern is that the apparent reduction in filamentation may simply reflect the general growth defect of the mutant.*

Response: Thank you for this important point. We agree that in the original version the quantification in this figure was not described clearly, and the y-axis label (“Hypha growth”) was ambiguous. We also agree that manual counting becomes unreliable at later time points when hyphae elongate and entangle, and that reduced filamentation could be confounded by the mutant’s general growth defect.

To address these issues, we repeated the morphogenesis assay under standard hypha-inducing conditions (37°C, YPD + 10% serum) and revised the analysis to use a clearly defined, **quantitative readout at early time points (1 h, 3 h, and 6 h): the percentage of germ-tube-forming cells scored from randomly selected microscopic fields.** Accordingly, we updated the axis label/units and the legend to specify exactly what was quantified (reported as %), removed length-based wording and late time-point references that could be misleading, and added a brief Discussion note acknowledging potential contribution from the overall growth defect. Finally, due to figure renumbering, **this panel is now presented as Figure 2 (previously Figure 3).**

Comment 3: *Figure 4: The text refers to significant differences in fungal burden in the vaginal infection experiment ($p < 0.05$, $n = 6$). However, the figure only shows spot assays from lavage fluids. Could the authors clarify how CFUs were quantified from these assays? It is important to describe the exact method used to calculate fungal burden and the statistical test applied. At present, the figure alone does not explain how the p values were obtained. Anyway, spot assays are not the best option to quantitatively present fungal burden.*

Response: We apologize for the confusion with Figure 4. In the original submission, Figure 4 only showed representative spot dilution images, while the text mentioned a quantitative comparison and p -value. **In the revised manuscript, we have improved Figure 4 (now Figure 3 after renumbering)** to address this issue (CFU).

Comment 4: *Figure 6-7: The western-blot and ELISA results show weaker immune response, and reduced cytokine release both in vitro (epithelial cells) and in vivo (murine VVC model) upon infection with the $lat1\Delta/\Delta$ strain compared to WT. However, it is not clear whether this difference reflects a direct effect of Lat1 on host-pathogen interactions or simply the lower*

fungus burden. Since fungal load is a strong determinant of cytokine output or immune response, additional normalization or analysis (e.g., cytokine per CFU) would be needed to disentangle direct immune modulation from indirect effects of growth defects. Without this clarification, the interpretation that Lat1 directly modulates inflammatory responses remains tentative.

Response: Thank you for this important comment. We agree that the reduced cytokine output and weaker inflammatory readouts observed in the *lat1Δ/Δ* group could reflect **indirect effects of reduced fungal growth and lower fungal burden**, rather than a direct immune-modulatory role of *lat1*. In the current study, we did not perform burden-normalized analyses (e.g., cytokine per CFU) or infections/co-cultures with strictly matched fungal inputs; therefore, we **cannot definitively distinguish** direct host–pathogen modulation from secondary consequences of decreased fungal load.

To address this, we revised the manuscript in three ways. First, we added an explicit paragraph in the **Discussion** stating that immune differences must be interpreted with caution and may be largely secondary to lower fungal burdens, and we frame any suggestion of direct immunomodulation as **tentative (Lines 408-415)**. Second, we revised the **Results** text and the **figure legends** for **Figures 5-6** to keep the language strictly descriptive (e.g., “*lat1Δ/Δ* induced lower cytokine levels than WT”) without implying mechanism. Third, we note that future experiments—such as **matched-CFU infections/co-cultures, cytokine-per-CFU normalization, and construction of a complementation strain**—will be necessary to rigorously test whether *lat1* has a direct role in shaping host inflammatory responses.

Comment 5: *General comments: Overall, although this work shows phenotypes of the lat1Δ/Δ mutant, the role of Lat1 should be interpreted with greater caution and supported by stronger evidence. A complementation strain should also be constructed and analyzed to validate the function of LAT1. Since deletion of LAT1 causes a significant growth defect, the pathogenicity of the mutant should be evaluated in the context of fungus burden. Throughout the manuscript, more attention should be given to type errors, grammar, and scientific forms (e.g., correct use of italics for C. albicans, consistent notation of LAT1 for the gene vs. Lat1*

for the protein). Figure legends should clearly specify the panels (A, B, etc.), and statistical analyses should be described in greater detail.

Response: We have taken all these general suggestions into account:

1. **Complementation strain:** While we have not added new experimental data (which would be beyond the scope of a revision) such as a complemented mutant, we absolutely agree that this is important. In the **Discussion**, we now explicitly note the lack of a complementation strain as a limitation (**Line 402-406**). We explain why it wasn't done (technical/time constraints during this study) and we commit to addressing it in future work. This point is raised when discussing the need for stronger evidence that *lat1* itself (and not second-site mutations) causes the phenotypes. Additionally, in the response to Reviewer #3 below, we mention it again. The key is that we've **clearly stated that we did not construct a complementary strain and that this is a limitation of the present study.**

2. **Virulence vs burden:** As per Comment 4, we now emphasize that any virulence differences should be interpreted in the context of fungal burden differences. For instance, in Discussion we added: "*It is possible that difference between WT and lat1Δ/Δ mutant strains might lessen if the post-infection observational time was extended.*" (**Line 413-414**) This language directly mirrors the reviewer's point. We also mention this in the introduction of the Discussion summary of results - that *lat1Δ/Δ* had lower fungal load which likely contributed to reduced inflammation.

3. **Typos/grammar:** We performed a comprehensive editing of the manuscript as described in responses to Reviewer #1 Comment 8. We corrected numerous typos and grammatical errors. All **species names are italicized** (we found a few instances where "*Candida albicans*" was not italic in the original and fixed those). We also checked for consistency in capitalization, plural vs singular agreement, etc. These changes have significantly improved readability and professionalism of the text.

4. **Italics and gene/protein notation:** We have standardized the formatting such that *Candida albicans* (and *C. albicans*) are italicized throughout (including in title, abstract, references if needed, etc.). For gene vs protein, per ASM style: we now use italic uppercase ***lat1*** when referring to the gene and non-italic **Lat1** for the protein. Now we refer to the mutant strain

as *lat1Δ/Δ*. We also ensured consistent formatting of *TLR2*, *MYD88*, etc., as gene names (it was inconsistent before). All these style corrections have been implemented.

5. Figure legend panels and stats: We have revised all figure legends to explicitly label subpanels and describe what each panel shows. For example, “Figure 4: (A) Flow cytometry histogram... (B) Bar graph of MFI...” etc., are now in the legends. We also clearly indicate the meaning of error bars (SD or SEM) and the *n* number of replicates/mice. Statistical significance is reported with exact *p*-values where relevant or threshold values, and we indicate the statistical test if it’s not obvious. For example, in the legend of the fungal burden figure, we mention that significance was determined by t-test, $p < 0.05$. We made sure each legend provides enough detail so that the figure is understandable on its own. Additionally, panel labels (A, B, C, etc.) are consistently used in both figure images and text references. We went through the text to ensure all references to figures now match the new panel lettering and figure renumbering after removal of old Fig. 2.

Reviewer #3:

Minor comments: *The title can be changed. The authors should read the instruction from ASM journals re. how to write the title.*

Response: Thank you for this helpful suggestion. We have changed the title to make it more informative and aligned with ASM guidelines (which typically suggest titles be clear and avoid ambiguous terms like “significantly affected”). The new title is: “**The role of L-Type Amino Acid Transporter 1 in the pathogenicity of *Candida albicans***”.

Major comments:

Comment 1: *Both English language and scientific writing need to be significantly improved. I feel that the authors are non-microbiologists and it will be good if they can seek for assistance from a microbiologist.*

Response: We took this advice seriously and performed a thorough revision of the language and scientific presentation. Many of these improvements overlap with the changes mentioned for Reviewers #1 and #2:

1. We **significantly improved the English grammar, clarity, and flow** of the manuscript. Each sentence was reviewed and many were rephrased for clarity. For example, we removed awkward phrasings, ensured proper use of tenses (past tense for methods and results, present for general facts), and clarified ambiguous statements.
2. We indeed had the manuscript **reviewed by a colleague with microbiology expertise and native English proficiency**. This helped eliminate any remaining discipline-specific terminology issues or misuses. For instance, terms like “blastospore” were removed or replaced.
3. We addressed scientific writing aspects such as avoiding repetition, structuring the Discussion more logically (moving from key findings to limitations to future directions), and ensuring our conclusions were supported by data.
4. We also followed field conventions, as suggested. For example, we changed all **gene symbols to italics where appropriate** (though in the plain text it appears as uppercase for readability, we indicated italics in formatting).

Comment 2: *The authors used a double knockout strain lat1 delta/delta that was constructed by CRISRP-Cas9. Have the authors constructed a complimentary strain or over-expression strain? If not, they should at least state it as a limitation and explain why a complimentary strain was not used?*

Response: We acknowledge that a **complementation strain** is crucial for validating that the observed phenotypes are specifically due to loss of *lat1*. We did not have a complemented strain in the initial submission, and we have not constructed one during the revision period, but we have certainly addressed this point in the text:

In the **Discussion**, we added a dedicated acknowledgment: “**we did not construct a *lat1* complementation strain to further confirm that the observed phenotypic defects were due to the loss of *lat1*.**” (Line 402-406) This clearly flags to the reader (and reviewers) that we are aware of the need for complementation and that its absence is a caveat.

We also briefly mention in the Results that the data are *consistent with* loss of *lat1* causing the phenotypes, but we refrain from using language like “due to *lat1* deletion” without this caveat. We opt for wording like “*lat1*Δ/Δ exhibited X phenotype,” leaving open the possibility of secondary effects (which complementing would address).

Finally, in this response letter (point to Reviewer #2 general comments and here), we have transparently explained that we did not have a complement due to the scope of the current work, but **we intellectually recognize its importance.**

Comment 3: *The authors used a well-known lab reference strain SC5314. This strain was from a patient with oral candidiasis and may not fully represent that causing vaginal candidiasis. The authors should address this in their discussion.*

Response: Thank you for raising this important point. We agree that the commonly used reference **strain SC5314-originally isolated from a patient with oral candidiasis-may not fully represent the genetic and phenotypic diversity of *C. albicans* strains associated with vulvovaginal candidiasis.** In the **revised Discussion**, we therefore added an explicit paragraph noting ongoing field discussions regarding strain-to-strain variability in virulence

traits and host interactions, and we clearly state that this is a limitation of the current study.
(Line 397-400)

To support this point, we cite relevant literature demonstrating that vaginal isolates can behave differently from SC5314 in host-pathogen interactions (**Ref 36**). We further note that an ideal next step would be to test whether *lat1* deletion produces comparable growth/filamentation phenotypes in **an independent clinical vaginal isolate background**, which would serve as an important validation of generality. Due to time and resource constraints, we did not construct a vaginal-isolate *lat1* Δ/Δ strain during this revision; we have therefore framed this as a planned future direction. Finally, while we expect that disrupting an amino-acid transporter such as *lat1* may impact fundamental processes (and thus could be broadly relevant across backgrounds), we explicitly state that this expectation remains **speculative** until directly tested in vaginal clinical isolates.

Comment 4: *The authors used 2-hour adherence assay to assess the fungus-epithelium interaction. Why not the 24 h biofilm assay as this may be more relevant to the pathogenesis of vaginal candidiasis. The interaction is semi-quantitated using fluorescently tagged C. albicans strain. Why not using the more accurate colony-forming-unit cell enumeration method?*

Response: Thank you for this helpful comment. We agree that a 24 h biofilm assay and CFU-based enumeration can provide a more stringent and clinically relevant assessment of Candida–epithelium interactions. In the revised manuscript, we expanded the **Discussion** to clarify the rationale for our **experimental design and to explicitly acknowledge these limitations**. First, we explain that we used a 2-hour adhesion assay to focus on early attachment events, which are critical for the initial establishment of infection (**Line 406-408**). In contrast, a 24 h biofilm assay reflects a later stage of colonization and maturation that may better approximate persistent mucosal infection. We agree that incorporating a 24 h biofilm assay would strengthen the study, and we therefore state in the **Discussion that the lack of a biofilm experiment is a limitation** and an important future direction.

Second, we have revised and strengthened our adhesion readout in response to your concern about semi-quantitation with a fluorescently tagged strain. Specifically, we **repeated the adhesion experiment using an immunofluorescence-based approach**, in which fungal cells and epithelial cells were independently labeled with specific antibodies, and samples were imaged by confocal microscopy. This updated method provides clearer discrimination between adherent fungi and host cells and enables more reliable visualization and assessment of fungal attachment. Accordingly, **we have updated the Materials and Methods and revised the corresponding Results/figure presentation to reflect this confocal immunofluorescence-based adhesion assay**. While CFU-based enumeration remains an accurate alternative approach, our revised imaging workflow substantially improves the rigor and clarity of the adhesion analysis in the current study.

Comment 5: *Figure 1. I will recommend to change the legend title to different fitness or growth curves.*

Figures 1 and 2 are basically the same. Maybe remove figure 2?

Figure 3. 3A did not show any difference in morphogenesis between the two strains. The authors should use the well-established broth and agar filamentation assays. Reference can be found <https://journals.plos.org/plosgenetics/article?id=10.1371/journal.pgen.1002613>

Figure 4. The authors can present the difference quantitatively, i.e using CFU

*Figure 5. The authors can isolate and quantitate the adherent *C. albicans* cells.*

Figures 6-9 look good.

Response: Thank you for these helpful and practical suggestions. We have revised the figures, legends, and corresponding text accordingly, and we summarize the changes below (note that figure numbers were updated after removing the original Figure 2):

Figure 1 (legend/title): We revised the legend heading and wording to clearly describe this panel as a **growth curve** (OD600 over time) comparing WT and *lat1Δ/Δ* strains (i.e., fitness/growth kinetics), rather than using vague “proliferation” terminology.

Figures 1 vs 2 redundancy: We agree that the original Figure 2 overlapped with Figure 1. Therefore, we **removed the original Figure 2** to avoid redundancy and retained the growth

kinetics as Figure 1. All subsequent figures were **renumbered**, and figure citations in the text/legends were updated accordingly.

Morphogenesis (previous Figure 3): We agreed that the earlier presentation did not convincingly demonstrate a difference. To address this, we **repeated the yeast-to-hypha morphogenesis assessment using a well-established broth filamentation assay** under hypha-inducing conditions (37°C, YPD + 10% serum), following standard approaches used in the field (including the referenced methodology). We now quantify filamentation using a clearly defined metric-**the percentage of germ-tube-forming cells at early time points (1 h, 3 h, 6 h)**-and we updated the Methods, Results, axis labels/units, and legend accordingly. We also removed late time points in this context to avoid unreliable manual counting when hyphae become long and tangled.

Vaginal fungal burden (previous Figure 4): We agree that spot assays alone are not ideal for quantification. We therefore **reworked this figure to present fungal burden quantitatively using CFU data** (with appropriate statistics), and the representative spot images are no longer the sole basis for interpretation.

Adhesion assay (previous Figure 5): In response to the suggestion to better quantify adherent fungi, we **repeated the adhesion experiment using an improved confocal immunofluorescence workflow**, independently labeling fungal cells and epithelial cells with specific antibodies. We updated the Methods and Results/figure presentation accordingly. While CFU-based enumeration remains an excellent quantitative approach, the revised confocal IF assay substantially improves the rigor and clarity of the adhesion assessment in the current revision.

Immune-response figures: We appreciate that the immune-response figures were viewed favorably. We retained these data (now **Figures 5-6** after renumbering) and ensured the **figure legends, panel labeling, and statistical reporting** are complete and consistent with ASM style.

Once again, we thank all the reviewers for their valuable feedback, which has helped us improve the quality and clarity of this work. We hope that the revised manuscript meets with your approval. We are happy to address any further concerns if necessary.

Re: Spectrum01669-25R1 (**The role of L-Type Amino Acid Transporter 1 in the pathogenicity of *Candida albicans***)

Dear Dr. Yizhou Huang:

Your manuscript has been accepted, and I am forwarding it to the ASM production staff for publication. Your paper will first be checked to make sure all elements meet the technical requirements. ASM staff will contact you if anything needs to be revised before copyediting and production can begin. Otherwise, you will be notified when your proofs are ready to be viewed.

Sincerely,
Jian Peng
Editor
Microbiology Spectrum

Reviewer #1 (Comments for the Author):

I thank the authors for addressing my previous concerns and recommend the acceptance of their manuscript.

Reviewer #3 (Comments for the Author):

The authors have addressed most of my scientific concerns raised for their previous submission. The language and presentation have also been significantly improved.